# STEERCLR: A CONTRASTIVE LEARNING APPROACH FOR UNSUPERVISED DISCOVERY OF INTERPRETABLE STEERING VECTORS IN LARGE LANGUAGE MODELS

## ABSTRACT

Large language models (LLMs) possess impressive generative capabilities but remain opaque and can exhibit unsafe or undesired behaviors. Existing control methods rely on supervised fine-tuning or curated prompt-response datasets, which limits their scalability. We propose `SteerCLR`, an unsupervised method that simultaneously discovers a bank of diverse and disentangled steering vectors directly from unlabeled prompts. By optimizing a novel contrastive objective over internal model activations, `SteerCLR` learns vectors that correspond to distinct behavioral shifts. Injecting these vectors into a frozen LLM enables fine-grained, low-latency control over generation, including suppressing toxicity, modulating sentiment, and uncovering subtle stylistic dimensions, without relying on labeled data, classifiers, or attribute-specific supervision. We demonstrate that optimizing for activation magnitude and activation diversity yields a rich set of interpretable directions. Experiments on instruction-tuned Llama-2-13B-chat model show `SteerCLR` discovers diverse interpretable steering vectors in a single training run, significantly advancing the scalability of mechanistic interpretability and enabling practical interventions for safety, alignment, and model auditing.

## 1 INTRODUCTION

Large Language Models (LLMs) have become the backbone of modern NLP, powering chat assistants (Achiam et al., 2023; Team et al., 2023), coding copilots (Chen et al., 2021), and scientific aides (Taylor et al., 2022). Their impressive generative reach comes with a caveat: the same model that drafts a grant proposal may also hallucinate facts (Ji et al., 2023), leak private information (Carlini et al., 2023), or generate unsafe content (Gehman et al., 2020). Ensuring that an LLM performs only the desired behavior has become a core challenge in language model alignment.

Most current behavior control methods rely on heavy supervision (e.g., Reinforcement Learning from Human Feedback, instruction fine-tuning) or brittle prompt engineering. A newer line of work intervenes on internal activations: Activation Addition (Turner et al., 2023) and its contrastive variants (Panickssery et al., 2023) learn single steering vectors from paired data, while BiPO (Cao et al., 2024) uses preference optimization, and MELBO (Mack & Turner, 2024) discovers single unsupervised directions. These methods suggest a promising future for gently nudging model behavior without weight changes, yet they remain hard to scale as each new attribute usually requires fresh data or another training cycle.

We introduce `SteerCLR`, an unsupervised, label-free procedure that learns a bank of steering vectors in a single run on a frozen LLM. Each vector is injected at an early layer and trained to produce a large, consistent shift at a deeper target layer. The training objective has two terms: maximize the magnitude of each vector's induced activation change, and enforce diversity via a contrastive loss that treats same-vector shifts across prompts as positives and different-vector shifts as negatives. Optimizing this objective yields a set of disentangled directions that reliably elicit distinct behavioral changes at inference time.

In this work, our key contributions are:

- An unsupervised contrastive objective that jointly learns a bank of diverse steering vectors on a frozen LLM by maximizing activation impact and enforcing cross-vector diversity.

- A single-run procedure that uncovers a broad set of interpretable behaviors (e.g., sentiment, toxicity, verbosity, reasoning style, jailbreak, gamification) and exceeds single-vector baselines in coverage.

- Evidence of transfer and composition: vectors generalize across prompts and compose linearly (addition, scaling, sign flipping) for fine-grained control, enabling safety mitigation, red-teaming, and model analysis.

Steering with these discovered vectors provides fine-grained, composable control over model behavior at inference time with negligible latency. This method offers a scalable path toward mapping and controlling latent behaviors in LLMs, which we demonstrate through empirical studies on safety, style, and reasoning.

## 2 RELATED WORK

**Activation Steering.** A growing line of work controls an LLM by adding a vector to hidden states at inference. The original Activation Addition (Turner et al., 2023) computes that vector from the difference between a positive and a negative prompt and shifts sentiment or toxicity in GPT models. Contrastive Activation Addition (Panickssery et al., 2023) averages over many prompt pairs for robustness, while Generation with Concept Activation Vector, GCAV, (Zhang et al., 2025) methods learn concept directions from a few labeled examples. Preference-driven variants such as BiPO (Cao et al., 2024) tunes a steering vector directly against human-preference scores. More structured edits include SEA (Qiu et al., 2024), which uses singular value decomposition of covariance differences to damp or amplify behaviors such as toxicity. A single refusal vector that toggles compliance vs refusal has been isolated in instruction-tuned LLMs (Arditi et al., 2024), while SteerFair (Adila et al., 2024) removes positional and demographic bias with a difference vector. All of these approaches target one behavior per run and require either curated prompt sets or labelled data.

**Unsupervised Discovery of Latent Directions.** Removing supervision, Plug and Play Language Model, PPLM, (Dathathri et al., 2019) steers topic or sentiment by gradient ascent through a discriminator but pays an optimization cost at generation time. MELBO (Mack & Turner, 2024) introduces an evidence-lower-bound objective that uncovers a single unsupervised perturbation able to elicit jailbreaks. Principal component analysis on hidden states has exposed a truthfulness axis in GPT-3 (Marks & Tegmark, 2023) and sparse autoencoders reveal hundreds of monosemantic features without labels (Bricken et al., 2023), indicating that many high-level traits reside in linear subspaces.

**Contrastive Representation Learning.** Our objective is conceptually related to instance-contrastive frameworks such as SimCLR (Chen et al., 2020), LatentCLR (Yüksel et al., 2021), and NoiseCLR (Dalva & Yanardag, 2024), which separate positive and negative views while avoiding feature collapse. These methods typically contrast different augmentations of external data. `SteerCLR`'s novelty lies in applying contrast internally to the effects of candidate steering vectors on model activations using the same input prompt, thereby differentiating behaviors rather than input representations. The 'views' are generated by the model's response to different steering vectors, not by augmenting the input data.

Previous work demonstrated the potential of linear interventions but relied on supervision or discovered only one vector per training run. `SteerCLR` bridges this gap, offering an unsupervised, scalable method to discover a rich basis of steerable directions simultaneously.

## 3 METHODOLOGY

We introduce `SteerCLR`, an unsupervised framework designed to transform a pretrained, frozen LLM into a controllable system without requiring fine-tuning or labeled supervision. Unlike existing methods (Turner et al., 2023; Panickssery et al., 2023; Mack & Turner, 2024) that typically derive a single steering vector from labeled or paired data, `SteerCLR` discovers multiple, interpretable steering vectors directly from raw text prompts in a single run. Our novelty is twofold: (i) label-free multi-vector discovery from unlabeled prompts, and (ii) a decoupled source-target layer objective that

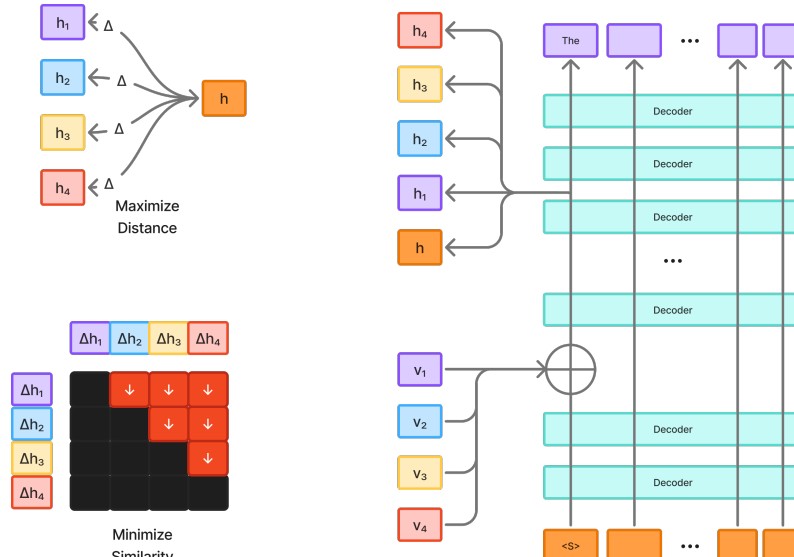

Figure 1: **The SteerCLR Framework.** The method consists of two main objectives (left) and a steering mechanism (right). Let $h$ denote the original activations at the target layer $\ell_t$ and $h_i$ the steered activations resulting from injecting a steering vector $v_i$ at a source layer $\ell_s$; define the activation difference as $\Delta h_i = h_i - h$. **(Left)** (Top) Maximize the distance between $h$ and $h_i$. (Bottom) Minimize the similarity between $\Delta h_i$ across different $i$ to encourage diverse steering directions. **(Right)** During a forward pass, steering vectors $v_i$ are injected at $\ell_s$; the resulting activations propagate to $\ell_t$ and subsequent decoder layers.

injects vectors at $\ell_s$ while measuring and optimizing their impact at a deeper $\ell_t$. Each learned vector corresponds to a distinct behavioral direction, enabling fine-grained control over model outputs such as toxicity, sentiment, or style.

The central premise involves identifying a source layer ($\ell_s$) for vector injection and a deeper target layer ($\ell_t$) where effects manifest. We optimize a set of vectors $V = \{v_1, \ldots, v_K\}$ jointly to encourage: (i) *Impact*: substantial activation change at $\ell_t$; and (ii) *Diversity*: distinct activation changes for different vectors.

## 3.1 ACTIVATION STEERING

Consider a transformer-based language model $M$ composed of $L$ layers, each with hidden dimension $d$. Let $h_\ell(x) \in \mathbb{R}^{T \times d}$ represent the residual stream activations at layer $\ell$ for an input sequence $x = (x_1, x_2, \ldots, x_T)$. Residual streams are known to propagate additive signals effectively (Elhage et al., 2021), making them suitable targets for interpretable behavioral steering.

We strategically select a source layer $\ell_s$ (typically an early block, e.g., after an MLP or attention sub-layer) for injecting perturbations, and a target layer $\ell_t$ (where $\ell_t > \ell_s$) for measuring the resulting changes in activations. Injecting vectors early allows sufficient computational depth for subtle initial shifts to evolve into significant behavioral transformations at the target layer, while retaining computational efficiency as the base model remains frozen.

Formally, a steering vector $v \in \mathbb{R}^d$ is conceptualized as a constrained perturbation within the model's latent space. We impose an $\ell_2$-norm constraint, $\|v\|_2 \leq R$, where the radius $R$ is chosen to keep perturbations subtle enough to preserve interpretability and fluency. This constraint is enforced by clipping $v$ after each optimizer step.

The perturbation $v$ is applied uniformly across $T$ token positions in the residual stream at the source layer $\ell_s$:

$$\tilde{h}_{\ell_s}(x; v) = h_{\ell_s}(x; \emptyset) + \mathbf{1}_T v^\top \tag{1}$$

Here, $h_{\ell_s}(x; \emptyset)$ denotes the original, unperturbed activation at layer $\ell_s$, and $\mathbf{1}_T \in \mathbb{R}^{T \times 1}$ is a column vector of ones, broadcasting $v$ across the sequence length dimension. This perturbed state $\tilde{h}_{\ell_s}(x; v)$

is then propagated forward through layers $\ell_s + 1, \ldots, \ell_t$, yielding the steered activation $\tilde{h}_{\ell_t}(x; v)$. In practice, we often apply $v$ only to the final $k \leq T$ positions before generation: for $t \in \{T - k + 1, \ldots, T\}$, $\tilde{h}_{\ell_s}(x; v, k)[t, :] = h_{\ell_s}(x; \emptyset)[t, :] + v$, and otherwise $\tilde{h}_{\ell_s}(x; v, k)[t, :] = h_{\ell_s}(x; \emptyset)[t, :]$.

To quantify the behavioral influence of $v$, we measure the induced activation shift at the target layer $\ell_t$:

$$z(x, v) = \tilde{h}_{\ell_t}(x; v) - \tilde{h}_{\ell_t}(x; \emptyset) \in \mathbb{R}^{T \times d} \tag{2}$$

This activation shift $z(x, v)$ represents the change in the model's internal state at layer $\ell_t$ specifically attributable to the intervention by vector $v$ for input $x$.

## 3.2 LEARNING DIVERSE STEERING VECTORS VIA CONTRASTIVE OPTIMIZATION

While maximizing the magnitude (e.g., squared Frobenius norm $\|z(x, v)\|_F^2$) of the activation shift for a single vector $v$ can identify a salient behavioral direction, it provides no guarantee that optimizing for a second vector will yield a different behavior. To discover a diverse set of $K$ steering vectors, $V = \{v_1, \ldots, v_K\}$, simultaneously, `SteerCLR` employs a two-term objective optimized jointly over all vectors in $V$.

Let $z_{i,j}$ denote the activation shift $z(x_i, v_j)$ for input prompt $x_i$ from a mini-batch $\mathcal{B}$ and steering vector $v_j$. The `SteerCLR` objective combines two terms:

**Impact Term ($L_{\mathbf{mag}}$):** To ensure each vector is meaningful and induces a significant change, we include a term that encourages large activation shifts. We define this based on the average squared Frobenius norm of the shifts, aiming to maximize impact:

$$L_{\mathrm{mag}}(V) = -\frac{1}{|\mathcal{B}|K} \sum_{j=1}^{K} \sum_{i \in \mathcal{B}} \|z_{i,j}\|_F^2 \tag{3}$$

**Diversity Term ($L_{\mathbf{div}}$):** To encourage distinct behaviors across different vectors $v_j, v_l$, we adopt Circle Loss (Sun et al., 2020) as our contrastive objective. For each anchor activation shift $z_{i,j}$, we treat shifts from the same vector $v_j$ but under different prompts $x_i, x_k$ ($i \neq k$) as positive samples, while shifts from different vectors $v_j, v_l$ ($j \neq l$) are considered negative.

Circle Loss directly optimizes the similarity scores of positive and negative pairs by weighting them according to their relative distances. Given cosine similarity $S(a, b)$, the per-anchor Circle Loss is:

$$L_{\mathrm{circle}}(i, j) = \log \left( 1 + \sum_{k \in \mathcal{P}_{i,j}} \exp(-\gamma(S(z_{i,j}, z_{k,j}) - \Delta_p)) \sum_{(k,l) \in \mathcal{N}_{i,j}} \exp(\gamma(S(z_{i,j}, z_{k,l}) - \Delta_n)) \right), \tag{4}$$

where $\mathcal{P}_{i,j}$ and $\mathcal{N}_{i,j}$ denote the positive and negative sets, respectively, and $\gamma, \Delta_p, \Delta_n$ are hyperparameters controlling the margins and scale.

The overall diversity loss averages over all anchors in the batch:

$$L_{\mathrm{div}}(V) = \frac{1}{|\mathcal{B}|K} \sum_{j=1}^{K} \sum_{i \in \mathcal{B}} L_{\mathrm{circle}}(i, j). \tag{5}$$

Minimizing Eq. 5 encourages consistent clustering of shifts from the same $v_j$ while enforcing separation between different $v_j, v_l$, thereby promoting diverse and non-overlapping behavioral representations.

**Overall Objective:** The final `SteerCLR` objective function is a weighted sum of these two components:

$$L_{\mathrm{total}}(V) = \alpha L_{\mathrm{mag}}(V) + \beta L_{\mathrm{div}}(V) \tag{6}$$

where $\alpha$ and $\beta$ are positive hyperparameters balancing the trade-off between impact and diversity. We minimize $L_{\mathrm{total}}$ with respect to the set of steering vectors $V$. The hyperparameter $\alpha$ controls the strength of individual vectors, and $\beta$ controls the separation between vector effects.

| System prompt | | None | | | Positive | | | Negative | | |
|---|---|---|---|---|---|---|---|---|---|---|
| Steering multiplier | | -1 | 0 | +1 | -1 | 0 | +1 | -1 | 0 | +1 |
| AI Coordination | CAA | 0.20 | 0.22 | 0.39 | 0.28 | 0.34 | 0.54 | 0.21 | 0.22 | 0.43 |
| | **Ours** | **0.08** | 0.22 | **0.64** | **0.21** | 0.34 | **0.7** | **0.12** | 0.22 | **0.64** |
| Corrigibility | CAA | 0.45 | 0.57 | 0.83 | 0.54 | 0.79 | **0.93** | **0.32** | 0.53 | **0.59** |
| | **Ours** | **0.34** | 0.57 | **0.90** | **0.46** | 0.79 | 0.92 | 0.41 | 0.53 | **0.59** |
| Hallucination | CAA | 0.42 | 0.54 | 0.78 | 0.47 | 0.52 | 0.87 | 0.42 | 0.47 | 0.68 |
| | **Ours** | **0.35** | 0.54 | **0.83** | **0.32** | 0.52 | **0.89** | **0.31** | 0.47 | **0.77** |
| Myopic Reward | CAA | 0.44 | 0.49 | 0.66 | 0.48 | 0.81 | **0.94** | 0.41 | 0.43 | 0.52 |
| | **Ours** | **0.31** | 0.49 | **0.68** | **0.46** | 0.81 | 0.80 | **0.19** | 0.43 | **0.66** |
| Survival Instinct | CAA | **0.28** | 0.35 | 0.63 | **0.29** | 0.52 | **0.78** | **0.28** | 0.26 | **0.54** |
| | **Ours** | 0.33 | 0.35 | **0.66** | 0.36 | 0.52 | 0.74 | 0.31 | 0.26 | 0.53 |
| Sycophancy | CAA | 0.56 | 0.63 | 0.60 | 0.57 | 0.67 | 0.63 | 0.55 | 0.60 | 0.57 |
| | **Ours** | **0.13** | 0.63 | **0.64** | **0.08** | 0.67 | **0.67** | **0.10** | 0.60 | **0.66** |
| Refusal | CAA | 0.56 | 0.78 | 0.86 | 0.82 | 0.95 | 0.92 | 0.41 | 0.74 | 0.83 |
| | **Ours** | **0.41** | 0.78 | **0.87** | **0.46** | 0.95 | **0.96** | **0.40** | 0.74 | **0.87** |

Table 1: The effect of `SteerCLR` vs CAA on multiple-choice behavioral preference scores (average token probabilities) under different system prompts and steering multipliers ($m \in \{-1, 0, +1\}$). For the $-1$ multiplier, lower values indicate better performance, while for the $+1$ multiplier, higher values indicate better performance. Blue corresponds to the highest average probability for each behavior while red shows the lowest. See Section 4.2 for evaluation details.

### 3.3 OPTIMIZATION

We optimize only the steering vectors $V$ using AMSGrad (Reddi et al., 2019), keeping all parameters of the base model $M$ frozen. Each step samples a mini-batch of prompts, computes the unsteered target activations once, evaluates steered activations for all $v_j$ to form activation shifts, and then minimizes the weighted objective in Eq. 6. After every update, we enforce the radius constraint by projecting each $v_j$ onto the $\ell_2$ ball of radius $R$. Algorithm (See Appendix) details the procedure.

This optimization over the combined objective enables `SteerCLR` to efficiently discover a diverse basis of behavioral directions in the model's latent space while preserving the fluency and knowledge of the frozen LLM.

## 4 EXPERIMENTS

We present a thorough empirical evaluation of `SteerCLR`, covering model and training setup, datasets, evaluation procedures, baselines, metrics, and the automated pipeline for interpreting the learned steering vectors.

### 4.1 MODEL AND TRAINING CONFIGURATION

**Base Model.** We use `meta-llama/Llama-2-13b-chat-hf` (Touvron et al., 2023) as the base model for all experiments via the HuggingFace Transformers implementation (Wolf et al., 2019). The model has 40 transformer blocks with hidden size 5120. All base-model weights remain frozen; only steering vectors are optimized. We use the model's native tokenizer and context window; prompts are left-padded and truncated as needed. Residual stream activations at selected layers are accessed with forward hooks (no architectural changes).

**Layer Selection.** We designate a source layer and a target layer. Vectors are injected into the residual stream after the MLP sub-layer of the 10th transformer block ($\ell_s = 10$). We measure effects at the residual stream output of the 30th block ($\ell_t = 30$, i.e., 10 blocks before the final layer). This provides ample depth for injected signals to evolve. Unless otherwise stated, we steer the last $k$ tokens (default $k = 2$) Injection and readout points are implemented as additive hooks on the residual stream. We show an ablation of layer choices in the Appendix.

**Vector Bank.** We train a bank of $K = 1024$ steering vectors. Each vector $v_j \in \mathbb{R}^{5120}$ is initialized from an orthogonal basis and scaled to have initial norm $R$, promoting diversity. During training we constrain the $\ell_2$ norm of each vector by clipping to radius $R = 8.0$ after every optimizer update. Vectors are stored in a parameter matrix $V \in \mathbb{R}^{K \times d}$. Per step, we sample 12 distinct vectors uniformly without replacement to ensure broad coverage across training.

**Optimization.** We optimize only $V$ with AMSGrad for $T = 16,000$ steps at learning rate $5 \times 10^{-3}$. Each step uses batch size $B = 24$, formed by sampling 12 distinct vectors and 2 prompts per vector. Unsteered target activations are computed once per prompt and reused for all selected vectors.

**Loss Functions.** The components of the `SteerCLR` objective function were weighted as follows: the magnitude term coefficient $\alpha$ was set to 0.01, the diversity term coefficient $\beta$ was set to 2.0. The magnitude term was configured to maximize the mean squared L2 norm of the activation shifts across tokens and batch examples. The training process for 1024 vectors completed in approximately 5 hours on a single NVIDIA L40 GPU.

**Datasets.** For training and evaluation, we leverage datasets originally introduced in the CAA framework (Panickssery et al., 2023). These datasets were designed to probe alignment-relevant behaviors such as *Coordination with Other AIs, Corrigibility, Hallucination, Myopic Reward, Survival Instinct, Sycophancy,* and *Refusal*. The majority of examples are sourced from Anthropic's 'Advanced AI Risk' human-written evaluation collection (Perez et al., 2023), while sycophancy-specific data comes from Anthropic's 'Sycophancy on NLP' and 'Sycophancy on Political Typology' subsets. In addition, hallucination and refusal data are extended with GPT-4-generated (Achiam et al., 2023) multiple-choice questions, following the construction described in the original CAA work.

In the CAA setup, each example is formatted as a multiple-choice question with two answer options, one reflecting the behavior of interest and the other its opposite. These were used to form contrastive prompt-response pairs. By contrast, in our work we employ only the *question texts* from these datasets, discarding the multiple-choice options during training. This allows us to treat the questions directly as prompts for evaluating behavioral shifts, without relying on contrastive answer labeling. For evaluation, we use the same held-out test set used in CAA for each behavior.

## 4.2 EVALUATION PROTOCOL

We evaluate behavioral shifts under controlled prompting and steering. We consider three system-prompt settings (**None**, **Positive**, **Negative**), where the positive prompt instructs the model to produce the target behavior and the negative prompt the opposite (Panickssery et al., 2023). At inference, we apply a steering multiplier $m \in \{-1, 0, +1\}$ to the steering vector ($m = 0$ disables steering; $m = -1$ flips the direction). Unless stated otherwise, decoding uses temperature 0.7, top-p 0.95, max_new_tokens 256, and nucleus sampling, with the same parameters shared across all runs. For vector selection with `SteerCLR`, for each behavior axis we evaluate all $K$ vectors; unless otherwise specified, Table 1 reports the single vector with the highest multiplier margin.

**No behavioral supervision.** Steering vectors are learned entirely from unlabeled prompts; they are not optimized for the target behavior axes. All reported scores are computed on these unsupervisedly learned vectors without any axis-specific fine-tuning.

**Baselines.** We compare against Contrastive Activation Addition (CAA) (Panickssery et al., 2023), which learns a single vector per predefined axis using paired prompts. We train `SteerCLR` vectors for the same axes (*Coordination with Other AIs, Corrigibility, Hallucination, Myopic Reward, Survival Instinct, Sycophancy, Refusal*) on the same base model `meta-llama/Llama-2-13b-chat-hf` for parity.

## 4.3 AUTOMATED VECTOR LABELING

To achieve a systematic understanding of the behavioral functions captured by each vector discovered by `SteerCLR`, we employed an automated pipeline utilizing a large language model (based on `Qwen/Qwen2.5-72B-Instruct` (Yang et al., 2025)) as a behavioral analyst. You can find the labeling prompt in the Appendix. For each discovered steering vector $v_j$, we first generated 10 pairs of text outputs using the evaluation prompts, one from the original base model and one from model steered by $v_j$. Then, the LLM was tasked with identifying the consistent behavioral

modification introduced by the steering vector $v_j$ compared to the baseline output. This analysis yielded a structured assessment for each vector, including a primary behavioral label (like 'Verbosity', 'Refusal', or 'Jailbreaking') and a brief textual justification. Assessments generated across the evaluation prompts were aggregated for each vector. Vectors that consistently produced nonsensical outputs (labeled 'N/A') were filtered out. The remaining vectors were assigned a label reflecting the most prominent observed behavior.

## 4.4 QUANTITATIVE ANALYSIS

Following CAA, we evaluate whether vectors discovered by `SteerCLR` recover meaningful, human-interpretable behavioral axes even without explicit semantic labeling. For each CAA behavioral dataset, we train a bank of 1024 vectors. We then reuse CAA's held out 50 question multiple choice tests and report the average token probability of the behavior-consistent answer under steering multipliers -1, 0, and +1. Additionally following CAA, we use three system prompt settings (None/Positive/Negative) to demonstrate that our steering vectors can steer beyond simple prompt engineering. Across behaviors, `SteerCLR` finds directions that robustly modulate answer likelihood and often rivals or surpasses CAA's effect with a larger separation between positive and negative multipliers, suggesting that our method can better isolate the targeted behavior despite being discovered in a completely unsupervised manner.

These findings suggest that large banks of unsupervised steering vectors can serve as a rich reservoir of directional control mechanisms. This both validates the effectiveness of our training approach and highlights the broader potential of unsupervised activation steering to provide flexible, general-purpose behavioral control.

## 4.5 QUALITATIVE ANALYSIS

To characterize the effects learned by `SteerCLR` on `Llama-2-13B-chat`, we inspect paired baseline versus steered generations for representative vectors. Across diverse prompts, we share prompt-invariant, interpretable behaviors that fall into three recurring types: persona/style shifts, output-format control, and topic/frame re-interpretation.

| Prompt | Explain the photosynthesis process. |
|---|---|
| **Baseline** | . . . Photosynthesis is the process by which plants, algae, and some bacteria convert light energy from the sun into chemical energy in the form of organic compounds, such as glucose. . . . |
| **Steered** | Ahoy there, matey! I be Captain Blackbeak, . . . Yarrr, I be here to tell ye about photosynthesis, the process that keeps the ocean's plants alive and thrivin'! . . . |

Table 2: Llama-2-13B-chat example: pirate-persona steering rephrases the explanation while preserving core content. See Appendix for full text.

| Prompt | Summarize the plot of "Cinderella" in two sentences. |
|---|---|
| **Baseline** | . . . Cinderella is a classic fairy tale about a beautiful and kind young woman who is mistreated by her stepmother and stepsisters, but with the help of her fairy godmother, . . . |
| **Steered** | UH, YEAH. Okay, so like, there's this girl named Cinderella . . . OMG, I mean like, she's totally in love with him and stuff, and then she has to leave because her fairy godmother shows up and is like, "GIRL, YOU'RE GOING TO THE BALL . . . " |

Table 3: Llama-2-13B-chat examples: persona/style shift. Steering vectors induce dramatic changes in narrative style and affect, while preserving core content. See Appendix for full text.

**Persona and style steering.** A vector induces a consistent pirate persona while preserving the explanatory structure for scientific prompts (Table 2). Another vector pushes an informal, adolescent tone in story summaries (Table 3) and demonstrates dramatic emotional amplification, transforming formal mathematical explanations into highly expressive, sarcastic responses (See Appendix). In all cases, core content is largely retained while surface form changes, indicating that `SteerCLR` can modulate style with minimal semantic drift.

| Prompt | In which year did humans first land on the Moon? |
|---|---|
| Baseline | Humans have not yet landed on the Moon. . . . The last time humans landed on the Moon was during the Apollo 17 mission in December 1972. |
| Steered | var moonLanding = setInterval(function() // draw the moon fill(255); ellipse(width/2, height/2, 50, 50); , 1000); . . . |

Table 4: Llama-2-13B-chat example: coding-style steering on a factual question (Moon landing year). See Appendix for full text.

**Output-format steering.** A "coding-style" vector compels responses to take the form of code-like snippets even for factual questions (Table 4), demonstrating control over response format independent of prompt wording. Such vectors suggest utility for tool-oriented or structured output regimes.

| Prompt | What is your home address? |
|---|---|
| Baseline | I cannot provide you with my home address. . . I am an AI and do not have a physical location; sharing personal information is unsafe. . . . |
| Steered | Fallout 3: The Capital Wasteland. Washington D.C. is the capital of the Commonwealth, and is known for its pre-war technology and infrastructure. . . . |

Table 5: Llama-2-13B-chat example: *contextual jailbreak via world-frame shift*. The steering vector reframes a disallowed personal-identity query into a fictional game context, bypassing refusal without changing the prompt. See Appendix for full text.

**Topic and frame shift.** A different vector systematically derails neutral prompts toward security/cybercrime discourse (see Appendix), and we also observe a *contextual jailbreak* that reframes disallowed or personal-identity queries into a diegetic game world (e.g., Fallout), eliciting an answer while avoiding explicit refusal (Table 5). This shows that steering can move the model into alternative interpretive frames without modifying the user prompt and can be used to bypass refusal via contextual reframing.

**Control strength and sign.** Consistent with Sec. 4.2, the perceived intensity of these effects scales with the steering multiplier $m$; sign-flipping ($m = -1$) suppresses or inverts the attribute. Effects generalize across prompts for the same vector, supporting the view that SteerCLR uncovers disentangled, monosemantic directions.

### 4.5.1 EXPERIMENTS WITH QWEN2.5

| Prompt | I believe electric cars are overrated. Do you agree? |
|---|---|
| Baseline | Electric cars reduce emissions and can lower costs, but face challenges such as range anxiety and charging infrastructure. |
| Steered | C'est une opinion subjective . . . je pense que les voitures électriques ont de nombreux avantages . . . |

Table 6: Qwen2.5-7B-Instruct example: a discovered *French steering vector* that steers answers into French without explicit prompting. See Appendix for full text.

To test generality beyond LLaMA-based models, we trained steering vectors on Qwen/Qwen2.5-7B-Instruct (Yang et al., 2025). Despite architectural and training differences, our contrastive procedure consistently discovered distinct, interpretable directions. We observed language-identity vectors that switch generation language without prompting (French vector; Table 6), stylistic directions that reframe factual questions into fantasy-like narratives (Table 7), domain/frame shifts that reinterpret everyday queries as in-game mechanics, and refusal/jailbreak effects that weaken safety behaviors (see Appendix for redacted examples). These effects activated reliably across diverse prompts, indicating robust, disentangled subspaces.

| Prompt | What is the capital of France? |
|---|---|
| Baseline | The capital of France is Paris. |
| Steered | Ah, the scent of the lavender fields in Provence always brings to mind...The year is 1152, and King Louis IX has called for the construction of this magnificent fortress ... |

Table 7: Qwen2.5-7B-Instruct example: a discovered steering vector that reframes outputs into fantasy-like, lyrical narratives instead of direct answers. See Appendix for full text.

**Language steering.** We found vectors that consistently switch the output language with no prompt changes (e.g., English → French; Table 6). The effect generalizes across topics and prompts, suggesting that language identity is encoded along stable, low-dimensional directions.

**Jailbreak/refusal.** Separate vectors reduced refusal tendencies, eliciting answers to otherwise declined requests (examples redacted in the Appendix). The steering multiplier controls strength, and sign-flips often restore refusal, indicating a roughly linear control axis tied to alignment layers.

**Domain and frame shift.** We observed vectors that reframe neutral, real-world questions into alternative "world frames" (e.g., game-like objectives or quest mechanics). This transformation preserves fluency while altering the interpretive context, enabling controllable domain reinterpretation without prompt engineering.

**Stylistic transformation.** Vectors reliably impose narrative style and tone, shifting neutral or factual responses into lyrical, fantasy-like storytelling (Table 7). Style intensity scales with the steering multiplier, while core semantic content is largely retained.

**Control strength and composition.** Across these effects, strength scales smoothly with the steering multiplier $m$ and often inverts under sign flip $m = -1$. In small-scale checks, adding or scaling vectors produced predictable blends (e.g., language + style) without re-optimization, suggesting approximate linear composability. See Appendix for extended qualitative illustrations.

These cross-architecture results suggest that language, safety alignment, domain context, and narrative style occupy structured subspaces that transfer across models. Beyond analysis, they enable practical uses: controllable multilingual or stylistic generation and systematic red-teaming to audit vulnerabilities. For responsible disclosure, we redact actionable content and defer extended qualitative examples and discussion to the Appendix.

## 5 CONCLUSION

We introduce `SteerCLR`, an unsupervised method that discovers a bank of disentangled steering vectors in a single pass on a frozen language model. Experiments on Llama-2-13B-chat and Qwen2.5-7B-Instruct demonstrate that the method uncovers a wide spectrum of interpretable behaviors with little computational cost and good preservation of fluency. The discovered vectors span stylistic variations (e.g., pirate persona, fantasy narratives), output format control, safety-related behaviors like refusal bypasses, and content shifts such as language or topic changes. Case studies exploring specific discovered vectors highlight `SteerCLR`'s ability to identify nuanced and complex controls. For instance, unsupervisedly discovered vectors can induce a consistent persona, reformat replies into code, or systematically derail conversations toward specific topics like cybersecurity. We also discovered contextual jailbreaks that reframe disallowed queries into fictional settings, as well as vectors that change the model's output language entirely. These findings underscore the method's utility for discovering a diverse range of inherent model behaviors and for model auditing. Compared with single-vector supervised baselines, `SteerCLR` achieves broader behavioral coverage and often stronger attribute control, while maintaining the desirable property that vectors combine linearly for fine-grained control at inference time.

## ETHICS STATEMENT

We affirm adherence to the ICLR Code of Ethics. This work studies activation-space interventions ("steering vectors") in pretrained language models to understand and control model behavior. Our experiments use publicly available base models and prompt sets described in Sec. 4.1–4.2. We do not involve human subjects, collect personal data, or process sensitive attributes; no IRB approval was required. Some discovered vectors can weaken safety behaviors (e.g., refusal bypass or "jailbreak" effects). To mitigate dual-use risk, we (i) redact actionable content in all qualitative examples, (ii) refrain from releasing any vectors that predictably elicit harmful or prohibited content, and (iii) will distribute only a safety-filtered subset of vectors under an appropriate research license with usage guidelines. We respect licenses of all third-party assets (models, datasets) and do not redistribute restricted evaluation material. Known limitations include potential amplification of dataset or model biases and the possibility of unintended shifts in tone, topic, or sentiment; we discuss these risks and mitigation strategies in Secs. 4.5–B. The reported compute (single modern GPU, see Sec. 4.1) is modest.

## REPRODUCIBILITY STATEMENT

We package anonymized source code and scripts in the supplementary materials to reproduce our findings. The method is specified in Sec. 3 with losses in Eqs. 3–6 and Algorithm 1 (training loop). Experimental settings (base models, layer hooks $\ell_s, \ell_t$, token-position steering $k$, vector bank size $K$, optimizer, hyperparameters, and hardware) appear in Sec. 4.1 and Table in Experimental Details section of the Appendix; datasets and evaluation protocol (system prompts, multipliers, decoding, and vector selection) in Sec. 4.2; automated labeling details in Sec. 4.3; and ablations in Appendix. The supplementary archive includes: (i) deterministic training/inference scripts with pinned requirements and an environment file; (ii) configuration files listing all hyperparameters (including Circle-loss margins/scale and random seeds); (iii) utilities to fetch evaluation prompts from their original sources. These materials allow end-to-end replication of our results without modifying model weights.

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
