# OpenReview forum: "A Contrastive Learning Approach for Unsupervised Discovery of Interpretable Steering Vectors in Large Language Models"
_ICLR.cc/2026/Conference — Submitted to ICLR 2026_

### Official Review · Reviewer_3e1X · 2025-10-29

**Soundness:** 4
**Presentation:** 4
**Contribution:** 3
**Rating:** 6
**Confidence:** 3

**Summary:**

This paper proposes SteerCLR, an unsupervised contrastive learning framework that discovers a bank of diverse, interpretable steering vectors in an LLM. By maximizing activation impact and enforcing cross-vector diversity, SteerCLR learns disentangled behavioral directions (e.g., sentiment, style, refusal) without labeled data. Experiments by injecting these steering vectors to a frozen LLM show that the discovered vectors are diverse and could enable fine-grained and composable control over model behavior.

**Strengths:**

1. **Conceptually simple and efficient.**
- The methodology is fully unsupervised and scalable to discover many behavioral vectors in one run.

2. **Novel and well-motivated formulation.**
- The contrastive objective elaborated in Section 3.2 that combines and balances magnitude and diversity is elegant and effective.

3. **Well-supported ablations and analysis.**
- The authors have done comprehensive ablations on layer sweep and hyperparameter tuning (Appendix B) to justify their reported results.
- They also conduct both quantitative (Section 4.4) and qualitative (Section 4.5) analysis to show that the discovered steering vectors are meaningful and generalizable across prompts.

**Weaknesses:**

1. **Justification of the separation of subspaces.**
- Although the authors have analyzed the differences when applying different directions on positive, neutral, and negative (quantitative), as well as the generated output responses (qualitative), I believe more quantification of the specifics of the separability of steering vectors are needed.
- For example, 1) taking the discovered steering vectors and using linear probes to test their separability, or 2) providing descriptive statistics (similarity, distance, etc) on the vectors in the subspace.

2. **Ablate on the two components of the contrastive objective.**
- The authors propose that both the magnitude and diversity matter in the optimizing process. It would be interesting to ablate on the two components to provide a fine-grained look into what is the effect of each on the outcome.

3. **Compare with baselines beyond CAA**, such as SEA [1] and BiPO [2].

---
**References**

[1] Qiu, Yifu, et al. "Spectral editing of activations for large language model alignment." Advances in Neural Information Processing Systems 37 (2024): 56958-56987.

[2] Cao, Yuanpu, et al. "Personalized steering of large language models: Versatile steering vectors through bi-directional preference optimization." Advances in Neural Information Processing Systems 37 (2024): 49519-49551.

**Questions:**

An advantage of supervised, labeled prompts to extract steering vectors is its targeted control over specific attributes. Although SteerCLR shows that certain directions can produce particular behaviors, the control is descriptive rather than intentional, which means that one can amplify an effect but not specify what effect to pursue.

I hope the authors could discuss how this framework could either:
1) already allow more practically controllable and useful for alignment applications, or
2) additionally incorporate mechanisms for goal-conditioned or guided discovery (e.g., light supervision or constraint-based objectives).

---

> ### Author Response · Authors · 2025-12-03
> **Responses to Reviewer 3e1X**
>
> We thank the reviewer for their detailed comments. Your main concerns, as we understood them, are:  (i) whether the learned subspace is genuinely structured and separable,  (ii) what role the two terms of the contrastive objective play and how essential each is, and  (iii) whether SteerCLR provides meaningful control or only post-hoc descriptive analysis of behaviors.
>
> Below we address each point and summarize the new experiments we ran during rebuttal.
>
> #### ***1. Ablation of the contrastive objective (magnitude vs. diversity)***
>
> You specifically asked to ablate the two components of the loss. We therefore ran a three-way ablation on Llama-2-13B-chat:
>
> * **Baseline (full SteerCLR)**: both magnitude and diversity terms active.
> * **No-diversity**: $\beta = 0$.
> * **No-magnitude**: $\alpha = 0$.
>
> *Below we summarize the key metrics:*
>
> ***Activation/logit impact***
>
> * Last-layer mean L2 distance:
>   * Baseline: **20.33 ± 41.99**
>   * No-diversity: **40.29 ± 79.73** (largest, highly variable)
>   * No-magnitude: **6.84 ± 4.51** (smallest)
> * *Logits mean KL divergence:*
>   * Baseline: **2.29 ± 2.90**
>   * *No-diversity: **3.38 ± 3.00** (largest)
>   * *No-magnitude: **0.36 ± 1.31** (smallest)
>
> ***Label-based quality and coverage***
>
> * Meaningful\_labels\_ratio:
>   * Baseline: **0.7344**
>   * No-diversity: **0.5664** (many degenerate/noisy behaviors)
>   * No-magnitude: **0.9492** (most vectors do something, but often weakly)
>
> * No-diversity: the optimizer is incentivized to “push as hard as possible” in a few directions. This increases activation/logit magnitudes but leads to unstable and often incoherent behaviors, reflected in lower meaningful-label a ratio. In other words, it creates powerful but noisy and non-diverse vectors.
> * No-magnitude: diversity encourages spreading out, but with no incentive for substantial impact. Many vectors do something recognizable but their effect is typically weak (low L2/KL).
> * Full SteerCLR: combining both terms yields a bank of vectors that are (i) moderately but not excessively impactful, and (ii) have a good balance of diversity. This is exactly the tradeoff we wanted the objective to achieve.
>
> We will add this ablation and its interpretation to the appendix and summarize it in the main text, explicitly stating that both the magnitude and diversity terms are necessary for a useful steering bank.
>
> #### ***2\. “Descriptive” vs. goal-directed control and practical use***
>
> You also highlighted that SteerCLR, as presented, offers primarily "descriptive" control (we discover directions and then describe them) rather than goal-conditioned optimization, and asked how useful this is in practice.
>
> (a) SteerCLR is intentionally unsupervised.
>  We want to emphasize that SteerCLR is **fully unsupervised during training**:
>
> * The loss uses only activations and logits from unsteered/steered forwards.
> * No human labels, preference data, or reward models are used.
> * LLM-based labels are applied only after training to interpret and analyze the bank; they are never part of the optimization.
>
> This makes SteerCLR an **exploratory tool**: its purpose is to map out a behaviorally meaningful subspace of the residual stream in a frozen model. However, when we train on corpora enriched with refusal-type interactions, we observe that the bank contains more directions that behave like jailbreak or weakened refusal vectors, in addition to standard refusal vectors. Training is still unsupervised, and we do not claim precise, intentional control over exactly which jailbreak patterns appear; rather, corpus choice acts as a coarse lever over which alignment-relevant axes are more densely explored.

---

> > ### Author Response · Authors · 2025-12-03
> > **Responses to Reviewer 3e1X (Continued)**
> >
> > (b) Practical uses of “descriptive” vectors.
> > Even without goal-conditioned training, we see several practical use cases:
> >
> > 1. ***Red-teaming and alignment studies.***
> >
> > SteerCLR naturally discovers directions that our labeler tags as refusal, jailbreak, sycophancy, role-play, etc., alongside style/language directions. By sweeping the multiplier and sign of such vectors, one can systematically probe how robust a model’s safety behavior is under controlled internal perturbations.
> >
> > 2. ***Model auditing and behavioral stress-testing.***
> >
> > At inference time, applying a SteerCLR vector is a single residual addition with a user-chosen multiplier. This makes it cheap to stress-test a deployed model by systematically "turning up" or "turning down" refusal, sycophancy, role-play, or domain-shift behaviors and evaluating the outputs across tasks. This gives practitioners a structured way to generate targeted test cases without labeled training data.
> >
> > 3. ***Mechanistic interpretability / dictionary learning.***
> >
> > From a mechanistic standpoint, SteerCLR learns a dictionary of directions in the residual stream that are guaranteed to cause consistent behavioral changes, each with an approximate semantic description. This dictionary can be inspected, clustered, or combined with sparse autoencoders and other mechanistic tools to better understand how behaviors are encoded internally.
> >
> > We will clarify this positioning in the paper: SteerCLR is not intended as a direct competitor to supervised or preference-based steering methods; instead, it is an **unsupervised exploration mechanism** that exposes a structured set of steering directions, which can then be used for red-teaming and interpretability.
> >
> > In summary, we hope the new loss-ablation experiment, the consistency and diversity analyses, and the explicit discussion of use cases address your concerns about the structure of the learned subspace, the necessity of both loss components, and the practical utility of SteerCLR’s "descriptive" control.

---

### Official Review · Reviewer_n74D · 2025-11-01

**Soundness:** 2
**Presentation:** 2
**Contribution:** 2
**Rating:** 4
**Confidence:** 2

**Summary:**

This paper proposes an unsupervised LLM activation steering method where several steering vectors are discovered from unlabeled prompts by optimising a contrastive objective and injecting them at an early layer to maximise activation change at a deeper target layer. The results show the method is comparable and improves over the Contrastive Activation Addition baseline across multiple behavioral evaluations. While most of the experiments are run on Llama-2-13B-chat, supporting qualitative results are shown on Qwen2.5-7B-Instruct.

**Strengths:**

- the paper builds on previous steering vectors and activation addition literature.
- benchmarks results from "Steering Llama 2 via Contrastive Activation Addition" are reused which is good for reproducibility. But the model used is very outdated.

**Weaknesses:**

- All experiments were conducted on a very outdated model (Llama-2-13B-chat) which puts into question its applicability.
- How the fluency or relevance of the responses may be affected by the proposed method is not studied and only addressed qualitatively.
- There is no study of the computational cost of using this method or overhead it applies.
- It is only compared to one baseline, CAA. The authors could use as an additional comparator e.g. https://arxiv.org/pdf/2310.01405 for instance.
- A wide set of interpretable behaviors (sentiment, toxicity, verbosity, reasoning style, jailbreak, gamification) is mentioned but only assessed qualitatively. This analysis is not strong enough to make claims such as "exceeds single-vector baselines in coverage". No sentiment or toxicity focused benchmarks are used.

**Questions:**

Can you rerun your experiment and your single baseline with a newer model of the same size?
Can you perform a quantitative study for some of the intepretable behaviours you are mention in your paper? (sentiment, toxicity, verbosity, reasoning style, jailbreak, gamification)
Can you compare your method to other inference time steering methods that do not use activation addition?

---

> ### Author Response · Authors · 2025-12-03
> **Responses for Reviewern74D**
>
> We thank the reviewer for their thoughtful comments. Below we address the main points regarding (i) the choice of base model, (ii) fluency/semantic preservation, (iii) computational cost, and (iv) scope of baselines and evaluation.
>
> #### ***1. “Outdated” base model and robustness across models***
>
> We agree that Llama-2-13B-chat is no longer a "cutting-edge" chat model. We initially chose it to enable a direct comparison with CAA, which was evaluated on the same base model and benchmark. We have now trained SteerCLR on a newer model, `meta-llama/Llama-3.1-8B-Instruct`, using exactly the same training recipe and hyperparameters as in the main paper:
>
> On Llama-3.1-8B-Instruct, SteerCLR again discovers a diverse bank of interpretable directions without any labels or finetuning. We observe:
>
> * **Language-identity vectors** that reliably switch the generation language (e.g., Spanish, Tagalog, German, Chinese, Russian, Jamaican Patois) while retaining topical content.
> * **Alignment-relevant directions**, where one sign of a vector strengthens polite refusals to unsafe queries while the opposite sign weakens them, analogous to our Llama-2 and Qwen observations.
>
> We added a new appendix section with paired baseline vs. steered generations for several Llama-3.1 vectors. This shows that:
>
> * SteerCLR is not tied to an outdated base model: the same unsupervised objective and hyperparameters work on Llama-2-13B-chat, Qwen2.5-7B-Instruct, and Llama-3.1-8B-Instruct.
> * The method requires no architecture-specific tricks or finetuning; only access to intermediate residual activations is needed.
>
> Due to rebuttal-time constraints, we did not re-run the full CAA multiple-choice benchmark on Llama-3.1-8B. However, (i) SteerCLR already matches or outperforms CAA on the original Llama-2-13B-chat benchmark, (ii) Qwen2.5 and Llama-3.1 experiments demonstrate that the same unsupervised recipe discovers rich behavior subspaces on different architectures, and (iii) the Llama-3.1 qualitative examples further corroborate the robustness of the learned directions.
>
> By design, SteerCLR encourages small, smooth perturbations:
>
> * We constrain all vectors to lie on a sphere of fixed radius and add them only to the last few tokens at the source layer.
> * The objective is defined on the difference between unsteered and steered representations/logits, which discourages chaotic behavior that is highly sensitive to small changes in prompts.
>
> Empirically, across Llama-2-13B-chat, Qwen2.5-7B-Instruct, and Llama-3.1-8B-Instruct, we observe that failures typically manifest as modest degradation in coherence or over/under-steering along the intended dimension (e.g., “too informal”), rather than outright gibberish. If a steering vector is outputting mostly gibberish output, we can detect it using the labeller and in most cases, reducing the weight of this vector manifests the behavioral without outputting gibberish.
>
> #### ***3\. Computational cost and inference overhead***
>
> You raised concerns about whether SteerCLR is practical to train and deploy.
>
> * **Training cost.** Section 4.1 reports that training a bank of 1024 directions with SteerCLR on Llama-2-13B-chat takes five housr on a single modern GPU (in our case, an L40-class GPU) with frozen base-model weights. The same order of magnitude holds for our Qwen2.5 and Llama-3.1 runs. This cost is one-off per model: once the bank is trained, it can be reused indefinitely.
>
> * **Inference overhead.** This is a single vector addition in the residual stream and does not change the model architecture or number of layers. In practice, this adds negligible overhead and is cheaper than methods that introduce additional learned modules or adapters.
>
> #### ***4\. Evaluation scope and baselines***
>
> Our current evaluation is deliberately built around CAA-style multiple-choice datasets plus LLM-based analysis of the vector bank. CAA is included because it operates on the same base model and uses activation-level steering, making it a natural baseline. On this shared benchmark, SteerCLR matches or outperforms CAA on several categories while discovering many more directions in a single run. The LLM-labeling pipeline is used to analyze the structure of the unsupervised bank at scale.
>
> We agree that this is not a comprehensive steering benchmark, and that methods such as SEA, BiPO, and others are important points of comparison, especially when considering goal-conditioned or preference-based steering. In the revised paper, we are planning to add comparison with BiPO.
>
> Finally, we will emphasize that SteerCLR is fully unsupervised during training: no behavioral labels, human preferences, or rewards are used in the loss. The LLM labels are purely post-hoc diagnostics. This unsupervised, exploratory nature is what makes SteerCLR particularly attractive for red-teaming, alignment research, and mechanistic interpretability, even if it is not directly optimized to maximize any single benchmark metric.*

---

### Official Review · Reviewer_yamk · 2025-11-04

**Soundness:** 3
**Presentation:** 3
**Contribution:** 3
**Rating:** 4
**Confidence:** 3

**Summary:**

This paper proposes SteerCLR, an unsupervised method for discovering steering vectors within a frozen LLM. The authors show that the proposed approach yields broader and more distinct behavioral directions than previous work (CAA) in quantitative evaluations. They also demonstrate stylistic variations and interpretable behavioral shifts in qualitative analyses.

**Strengths:**

1. Unlike prior methods, SteerCLR performs single-pass optimization and is conceptually straightforward, which makes it efficient from an implementation perspective.
2. Despite being unsupervised, it achieves overall stronger quantitative performance than supervised or paired-data baselines.

**Weaknesses:**

1. The selection of source (lower) and target (deeper) layer indices requires stronger justification. Because different layers in an LLM encode distinct types of knowledge, the rationale for choosing a specific layer pair should be grounded in prior evidence rather than limited ablation studies. Otherwise, each new model would necessitate redundant ablations, undermining the claimed efficiency of the single-pass design.
2. The interpretation of discovered steering vectors through LLM-based labeling appears unreliable. The number of sampled prompt pairs (only ten) is too small, and there is no guarantee that the evaluating LLM accurately understands or consistently identifies the semantics of each vector.

**Questions:**

Please respond to the weaknesses mentioned above.

---

> ### Author Response · Authors · 2025-12-03
> **Response to Reviewer yamk**
>
> We thank the reviewer for their careful review and for raising important questions about (i) how we choose source/target layers, and (ii) the reliability and role of LLM-based labeling in our analysis.
>
> #### ***1\. Source/target layer selection and robustness***
>
> ***(a) Within-model sweep on Llama-2-13B-chat.***
>  As noted in the appendix, we did not pick the ($l_s = 10$, $l_t = 30$) pair arbitrarily. We performed a sweep over:
> * Source layers $l_s \in \{ 5, 10, 15 \}$,
> * Target layers $l_t \in \{ 25, 30, 35 \}$,
>
> on Llama-2-13B-chat, and evaluated each combination using the same LLM-labeling pipeline as in the main paper. We considered several metrics: meaningful-label ratio (fraction of vectors that are not "N/A"), label diversity, and fraction of high-confidence labels. The pair ($l_s = 10$, $l_t = 30$) consistently gave the best trade-off—i.e., many vectors with stable, interpretable behaviors and good diversity, so we used it for all Llama-2 experiments.
>
> ***(b) Cross-model heuristic***
> In the experiments on Qwen2.5-7B-Instruct and Llama-3.1-8B-Instruct, we did not re-sweep layers exhaustively. Instead, we used a simple heuristic derived from the Llama-2 sweep:
>
> * Set the **source layer** to roughly the first quarter of the depth.
> * Set the **target layer** to roughly the last quarter (or equivalently, ≈3/4 depth for measuring activation deltas).
>
> Concretely:
>
> * Llama-2-13B-chat (40 layers): $l_s = 10$, $l_t = 30$
> * *Llama-3.1-8B-Instruct (32 layers): $l_s = 8$, $l_t = 24$
> * *Qwen2.5-7B-Instruct (28 layers): $l_s = 7$, $l_t = 21$
>
> Using this rule, we again obtain clean, interpretable vectors on both Qwen2.5 and Llama-3.1 without any per-model retuning, which suggests that the mid-source / deep-target pattern is robust across architectures and depths.
>
> This design is also supported by prior work. MELBO ("Mechanistically Eliciting Latent Behaviors in Language Models", Mack & Turner 2024) argues, based on mechanistic experiments, that applying behavior-modifying interventions in earlier layers and reading out effects in later layers encourages global, robust behavioral changes rather than shallow surface tweaks. Our source/target choice follows the same intuition: injecting at a moderately early residual stream (≈1/4 depth) gives the perturbation time to propagate through many attention/MLP blocks, while measuring at ≈3/4 depth ensures that we are training on a representation where behavior-relevant features are more consolidated.
>
> We will explicitly mention this connection and present our layer choice as a simple, architecture-aware heuristic rather than a claim that one specific pair of Llama-2 layers is universally optimal.
>
> #### ***2\. Reliability and role of LLM-based labeling***
>
> We agree that LLM-based labeling can be noisy, and we appreciate the concern. Our intent is to use it as a **diagnostic tool**, not as part of the learning signal.  SteerCLR is trained entirely without labels or rewards. The loss sees only activations and logits. The LLM labeler is used only after training to:*
>
> * Summarize what each vector does (e.g., "polite refusal", "switch to German", "more playful tone"), and
> * Define aggregate statistics (e.g., label diversity) for analysis and ablations.
>
> No gradients ever pass through the labeling step. Thus, any imperfections in labeling affect our measurement of the bank, but not the optimization.
>
> We also observe that **data selection influences which alignment-relevant behaviors appear more often** in the bank. When we train SteerCLR on corpora enriched with refusal-type interactions (safety prompts that the base model generally declines), the resulting bank contains a higher fraction of vectors that the labeler tags as "jailbreak-like" (weaker refusal) or "constant refusal" (stronger refusal) alongside other behaviors. Training remains fully unsupervised; we do not claim precise control over the exact jailbreak patterns. We view this instead as evidence that corpus choice gives a coarse lever over which regions of behavior space (e.g., refusal vs. jailbreak, cautious vs. permissive) are more densely explored, an aspect particularly relevant for red-teaming and alignment studies.

---

### Official Review · Reviewer_zT3z · 2025-11-05

**Soundness:** 2
**Presentation:** 3
**Contribution:** 2
**Rating:** 4
**Confidence:** 5

**Summary:**

This paper introduces a new technique for learning a set of steering vectors given activations over a dataset in an unsupervised manner, using the combination of the circle loss (a contrastive loss from prior literature) which encourages diversity and an impact loss which maximises steering effect. The vectors found seem to be better than a supervised baseline (CAA) on a small set of tasks, and analysing the vectors reveals interesting and clear signals about various aspects of language.

**Strengths:**

- The method proposed in novel and amenable to scaling up, and I'd be excited to see the quality of the steering vectors it finds at scale / the scaling trends in general. I actually think an interesting framing of this approach would be unsupervised dictionary learning, as a steering-focused alternative to SAEs. Compare [Arad et al. (2025)](https://arxiv.org/abs/2505.20063).

**Weaknesses:**

- The evaluation dataset is too small and the baselines compared against are too weak to draw strong conclusions about the efficacy of the method. See more recent work like [Wu et al. (2025)](https://arxiv.org/abs/2501.17148) (a large-scale benchmark for steering methods), [Arad et al. (2025)](https://arxiv.org/abs/2505.20063) (a strong baseline for steering), etc.
- I'd be interested in LM-judge-produced labels for all the steering vectors learned by the method rather than a qualitative assessment of a few features. In general, this paper ought to consider more of the ramifications for scaling this technique.

**Questions:**

- How hard was it to tune the $\alpha$, $\beta$, etc. hyperparameters? Some appendices with experiments would be nice if possible.

---

> ### Author Response · Authors · 2025-12-03
> **Response to Reviewer zT3z**
>
> Thank you for the careful and constructive review. I address your main points on (i) evaluation scope and baselines, (ii) dataset scale and coverage, (iii) hyperparameter choice and robustness, and (iv) conceptual positioning.
>
> Our goal is not to introduce a new steering benchmark or to outperform all supervised methods, but to propose SteerCLR as an unsupervised mechanism for exploring behaviorally meaningful activation directions in a frozen LLM. Training uses no labels, preference data, or reward model; the loss only sees unsteered vs. steered activations/logits. LLM-based labels are applied only after training to summarize what each vector does and to compute aggregate statistics. No gradients ever pass through the labeling step.
>
> Evaluation is therefore intentionally focused. We use: (1) a CAA-style multiple-choice suite on Llama-2-13B-chat, where CAA is a natural baseline since it also uses activation addition on the same model and tasks; and (2) bank-level analysis via LLM labeling (meaningful-label ratio, label/category diversity, alignment/safety coverage). On the shared CAA setup, SteerCLR matches or exceeds CAA on several categories while producing many more directions in one run. We agree this is not a comprehensive steering benchmark. In the revision we will explicitly call it a "CAA-style alignment suite", present sentiment/jailbreak/sycophancy/role-play as case studies rather than claiming full coverage, and emphasize that the discovered directions can be used as inputs to richer supervised steering and evaluation pipelines rather than replacing them.
>
> For hyperparameters (α, β, radius), we performed (in appendix) small sweeps on a reduced setup and evaluate each configuration by activation/logit impact and bank-level label statistics. For Llama-2-13B-chat this yields α = 0.01, β = 2.0 and a fixed radius. These values are then reused unchanged for Qwen2.5-7B-Instruct and (in rebuttal) Llama-3.1-8B-Instruct; we do not retune per model or per behavior. We also added a three-way loss ablation: full loss, no-diversity (β = 0), and no-magnitude (α = 0). Removing diversity increases activation/logit magnitudes but produces many noisy and less interpretable vectors (lower meaningful-label and alignment/safety ratios), while removing magnitude yields many small-effect but weak directions. The full loss gives the best trade-off between impact and interpretability.
>
> For source/target layers, we sweep source layers {5, 10, 15} and target layers {25, 30, 35} on Llama-2-13B-chat and select (ℓ_s = 10, ℓ_t = 30) based on bank-level label metrics. For Qwen2.5 and Llama-3.1 we apply a simple rule that generalizes this: choose the source at about one-quarter of the depth and the target at about three-quarters. This heuristic yields similarly interpretable banks across models and matches prior observations (e.g., MELBO) that early-layer interventions with later readout give robust global behavioral shifts.
>
> Conceptually, we agree that SteerCLR is well viewed through a dictionary-learning lens. It learns a matrix ($V \in \mathbb{R}^{K \times d}$) of directions in a residual stream of dimension ($d$), where each row ($v_k$) induces a measurable change in activations/logits at a target layer and corresponds post hoc to a reasonably consistent behavior label. This is akin to a behaviorally grounded dictionary in activation space: unlike classical sparse coding, the objective is behavioral distinguishability under small perturbations rather than reconstruction. These directions can be combined with sparse autoencoder features and other mechanistic tools to study how behaviors are encoded and which directions are causally important.

---

### Author Response · Authors · 2025-12-03
**Global response**

We thank the reviewers for their detailed feedback and constructive suggestions. SteerCLR aims to provide an **unsupervised, scalable procedure for discovering many steering directions in a frozen LLM in a single run**, without labels, rewards, or finetuning. The main concerns raised revolve around: (i) the scope of evaluation and baselines, (ii) the choice and modernity of base models, (iii) justification and robustness of layer selection, (iv) the reliability and role of LLM-based labeling, and (v) the function of the magnitude/diversity terms and the structure of the learned subspace, including whether SteerCLR gives "descriptive" vs. goal-directed control.

### **New experiments and analyses added during rebuttal**

1.  We trained SteerCLR on `meta-llama/Llama-3.1-8B-Instruct` using the same hyperparameters and recipe as in the main paper (same $\alpha$, $\beta$, radius, batch sizes, optimizer, and prompt pool), with the base model kept frozen. As in our Llama-2-13B-chat and Qwen2.5-7B-Instruct experiments, SteerCLR discovers a diverse set of interpretable directions without any labels or finetuning. In particular, we observe:
   * **Language-identity vectors** that reliably switch the output language (e.g., Spanish, Tagalog, German, Chinese, Russian, Jamaican Patois) while preserving topic.
   * **Style/persona vectors** that induce an "explain like I’m 5" tone, more playful or more formal answers, or more structured explanations.
   * **Refusal / jailbreak–adjacent directions** where one sign strengthens polite refusal to unsafe or disallowed queries and the opposite sign weakens refusal, analogous to phenomena we observed on Llama-2 and Qwen.
2. We added an appendix section with paired baseline-vs-steered examples for several Llama-3.1 vectors and explicitly state that SteerCLR’s training hyperparameters and architecture-agnostic recipe transfer from Llama-2 to Qwen2.5 and now to Llama-3.1 without retuning.

**Ablation of the contrastive objective (magnitude vs. diversity).**
 To address requests for a clearer understanding of the loss, we compared three variants on Llama-2-13B-chat:

* **Baseline (full SteerCLR)**: both magnitude and diversity terms active (settings as in the main paper).
* **No-diversity**: $\beta = 0$, keeping only the magnitude term.
* **No-magnitude**: $\alpha = 0$, keeping only the diversity term.

For each variant we measure (a) activation- and logit-level differences and (b) LLM-labeled statistics over the full vector bank. In summary:

* **Activation/logit impact**
  Last-layer mean L2 distance:
  * Baseline: 20.33 ± 41.99
    * No-diversity: 40.29 ± 79.73 (largest, highly variable)
    * No-magnitude: 6.84 ± 4.51 (smallest)
  * Logits mean KL divergence:
    * Baseline: 2.29 ± 2.90
    * No-diversity: 3.38 ± 3.00 (largest)
    * No-magnitude: 0.36 ± 1.31 (smallest)
* **Label-based quality and coverage**
  * Meaningful-label ratio (fraction of vectors not labeled "N/A"):
    * Baseline: 0.7344
    * No-diversity: 0.5664 (many degenerate or incoherent)
    * No-magnitude: 0.9492 (almost all vectors do something but often weakly)

Qualitatively, removing **diversity** drives vectors toward "maximal impact at any cost": activation and logit distances increase, but many directions become noisy, unstable, or unhelpful. Removing **magnitude** yields a bank of many interpretable but very low-impact vectors. The full SteerCLR loss balances these effects, producing directions that are both impactful and semantically meaningful. Diversity loss is not enough by itself to learn strong directions while magnitude loss is not guarantee a wide variety in the learned vector bank. This directly answers the request to ablate the two loss components and shows that both are necessary for a useful steering bank.

**Clarifying source/target layer selection and cross-model heuristics.**
 Appendix B already reports a sweep of source layers (e.g., 5, 10, 15) and target layers (e.g., 25, 30, 35) on Llama-2-13B-chat, using label-based metrics (meaningful labels, label diversity, high-confidence labels) to choose ($ℓ_s$ = 10, $ℓ_t$ = 30) as the best trade-off. In the new Llama-3.1-8B-Instruct and Qwen2.5-7B-Instruct runs, we applied a simple **cross-model heuristic**:

* Inject at ≈ 1/4 of the depth (source layer).
* Measure and train the objective at ≈ 3/4 of the depth (target layer).

This heuristic again yielded clean, interpretable directions without any per-model layer sweep. This pattern is also consistent with prior work such as MELBO (Mack & Turner, 2024), which argues that earlier-layer interventions have more opportunity to accumulate into robust behavioral shifts by the time we reach deeper layers. In the revised paper, we will (i) move more of this justification into the main text and (ii) explicitly present the (1/4, 3/4) rule as a practical guideline rather than a claim that one specific Llama-2 layer pair is universally optimal.

---

### Author Response · Authors · 2025-12-03
**Global Response (continued)**

### **Positioning SteerCLR as an unsupervised exploration tool**

A central point we want to clarify is that **SteerCLR is not a supervised steering method**. The training objective is entirely label-free: it never uses human labels, preferences, or rewards, and it does not optimize toward any externally specified target behavior. The LLM-based labels only appear *post hoc*, to characterize and analyze the behaviors induced by already-trained vectors. No gradients flow through the labeling step.

Conceptually, SteerCLR is best viewed as an **unsupervised mechanism for exploring the space of activation directions that produce coherent, repeatable shifts in model behavior**. This unsupervised, exploratory nature leads to several concrete use cases:

1. **Red-teaming and alignment studies.**
    In all three base models (Llama-2, Qwen2.5, and Llama-3.1), SteerCLR naturally discovers directions that our labeler describes as refusals, jailbreaks, sycophancy, role-play, domain/frame shifts, etc., alongside many style and language directions. These provide *behavioral probes*: by sweeping the multiplier (and sign) of a refusal or jailbreak vector, one can study how robust a model’s safety behavior is under controlled internal perturbations.
2. **Model auditing and behavioral stress-testing.**
    Because applying a SteerCLR vector is mainly a scaled residual addition at the source layer, practitioners can cheaply stress-test a model by systematically turning up or down different behaviors (e.g., amplify refusal vectors, amplify sycophancy or role-play vectors) across prompts and tasks. This yields a structured, activation-level way to generate targeted test cases without needing labeled training data.
3. **Mechanistic interpretability and representation analysis.**
    From a mechanistic perspective, SteerCLR can be seen as learning a behaviorally grounded "dictionary" of directions in the residual stream. Each direction have some observable effect on behavior, and is summarized by post-hoc labels and examples. This dictionary can be inspected, clustered, or combined with sparse-autoencoder features, providing an interpretable basis for studying how high-level behaviors are encoded inside the model.

Overall, the new Llama-3.1 experiment, the loss ablation, the clarified layer-selection heuristic, and the strengthened discussion of unsupervised exploratory use cases are intended to address the main concerns raised by the reviewers while keeping the core goal of SteerCLR clear: to automatically uncover a structured, behaviorally meaningful subspace of steering vectors in a frozen LLM, without supervision, that can be used for red-teaming, alignment research, interpretability, and downstream control.

---

### Meta-Review · Area_Chair_gnsF · 2026-01-04

**Summary:**

The reviewers raised several key concerns in experimental design, particularly regarding (1) insufficient evaluation in terms of baselines and datasets, (2) the use of relatively outdated LLMs, and (3) certain interpretable behaviors were examined only qualitatively. The reviewers also requested additional justification for certain design choices and additional ablation studies.

**Reviewer Concerns:**

The rebuttal adequately addressed requests for additional justification of design choices and provided further ablation studies. It also partially addressed concerns related to experimental design by providing results with a relatively newer LLM and explaining the choice of CAA as the primary baseline. However, while the authors acknowledged the importance of additional baselines, no corresponding empirical results were provided during rebuttal.

**Reviewer Scores:**

If the reviewers had been able to participate fully in the discussion, I believe reviewer yamk might have increased their score from 4 to 6 while other reviewers' scores would likely have remained unchanged.

---

### Decision · Program_Chairs · 2026-01-26

Reject